# Research on the impact of inclusive finance on agricultural green development: Empirical analysis of China's main grain producing areas

**Aihua Tong[1], Lili Jiang[1,2], Yufan Ru[3], Zhifei Hu[1,4], Zhongrong Xu[1], Yifeng Wang[1]\***

**1** Suqian University, Suqian, China, **2** China University of Mining and Technology, Xuzhou, China, **3** Fujian Agriculture and Forestry University, Fuzhou, China, **4** Shanghai University, Shanghai, China

\* wyf870403@163.com

**Data Availability Statement:** All relevant data are within the paper and its Supporting Information files.

## Abstract

In order to study the impact of inclusive finance on agricultural green development, This paper uses both static panel and dynamic panel (system GMM model) estimation methods to make empirical analysis of the impact of inclusive financial development on agricultural green development. The results both find that there is a significant positive correlation between the level of inclusive financial development, real GDP per capita, the proportion of the added value of agriculture, forestry, animal husbandry and fishery in GDP and agricultural green development. This paper puts forward some countermeasures and suggestions to promote agricultural green development, including vigorously developing inclusive finance, promoting economic growth, promoting the development of agriculture, forestry, animal husbandry and fishery, and increasing environmental protection expenditures.

## Introduction

Since the reform and opening up, China's agricultural development has made remarkable achievements. At the same time, it is also facing some problems such as the deterioration of the agricultural ecological environment, the excessive utilization of agricultural resources, and the excessive investment of agricultural materials such as fertilizers, pesticides, and agricultural films. Agricultural green development has gradually gained people's attention as a way of sustainable agricultural development to change the traditional oil agriculture. Agricultural green development is a sustainable agricultural development mode that aims to achieve the coordination and unity of economic, social and ecological benefits through the universal implementation of green agricultural production technologies and methods on the premise of protecting the environment of agricultural production resources. Agricultural green development is the direction of modern agricultural development, which is related to the harmonious development of man and nature, the quality and safety of agricultural products, and the sustainable development of agriculture. The *Opinions on Promoting Green Agricultural Development through Innovation Mechanisms* issued in 2017 called for accelerating green agricultural development. The report of the 19th National Congress of the Communist Party of China raised green agricultural development as a national strategy, *National Strategic Plan for Promoting*

**Funding:** This paper was supported by Research on the path and model of innovation-driven agricultural transformation in Jiangsu (20GLD014); 2020 General Project of Philosophy and Social Sciences Research in Jiangsu Universities - Research on Financial Support for Agricultural Green Development in Jiangsu Province (2020SJA2177).

**Competing interests:** The authors have declared that no competing interests exist.

*Agriculture by Quality (2018–2022)* proposes to focus on promoting agricultural green development. China's *14th Five-Year Plan* also emphasizes promoting the green transformation of agriculture. China's main grain producing areas include 13 provinces including Hebei, Inner Mongolia, Liaoning, Jilin, Heilongjiang, Jiangsu, Anhui, Jiangxi, Shandong, Henan, Hubei, Hunan, and Sichuan. The grain output of 13 provinces accounts for more than 70% of China's grain output. Improving the agricultural green development level of the 13 provinces in China's main grain-producing areas is of great significance for promoting the harmonious development of man and nature, for ensuring the safety of China's grain quantity and quality, and for meeting consumers' growing demand for high-quality agricultural products.

Promoting agricultural green development requires strong support from inclusive finance. Agricultural green development requires the effective allocation of various production factors such as land, capital, technology, and institutions. One of the key factors is capital, which plays a role in guiding the allocation of other production factors. It is necessary to effectively connect rural finance with the national agricultural green development policy to promote agricultural green development. Inclusive finance focuses on providing financial services to various groups of society and meeting the financial needs of different groups. Due to the inclusiveness and wide coverage of inclusive finance, inclusive finance plays a very important role in promoting agricultural green development. The development of inclusive finance can better meet the financial needs of agricultural green business entities, and can provide financial services such as funding services and insurance services for promoting agricultural green development. In 2018, the Opinions of the Central Committee of the Communist Party of China and the State Council on the Implementation of the Rural Revitalization Strategy emphasized that the development of inclusive finance should be focused on rural areas.

Under the background of financing constraints and high consumption of chemical products in agricultural green development, it is worth in-depth research to explore whether inclusive finance is conducive to promoting green agricultural development. It is particularly necessary for the government to formulate policies and allocate inclusive financial resources in the market.

Existing studies mainly focus on the impact of inclusive finance on urban and rural income, consumption, poverty reduction, employment, entrepreneurship and innovation. However, there are few studies on the relationship between inclusive finance and agricultural green development, and the empirical analysis of the impact of inclusive finance on agricultural green development. Therefore, whether inclusive finance affects agricultural green development and to what extent it affects agricultural green development requires further research. This paper focuses on the impact of inclusive finance on agricultural green development. This paper analyzes the development of inclusive finance, the development of green agriculture, and whether inclusive finance affects agricultural green development in 13 provinces in China's main grain producing areas.

The main contributions of this paper are as follows: Based on the theoretical analysis of the impact of inclusive finance on agricultural green development, this paper empirically analyzes the specific impact of inclusive finance on agricultural green development. By constructing a scientific evaluation index system for the development of inclusive finance, and using the coefficient of variation method to calculate Index of Financial Inclusion in 13 provinces from 2011 to 2019, this paper evaluates the development level of inclusive finance in 13 provinces; by constructing an evaluation index system for agricultural green development, using the entropy weight method to calculate the Agricultural Green Development Index of 13 provinces in China from 2011 to 2019, this paper evaluates the level of agricultural green development in 13 provinces. Using static panel (mixed least squares estimation (Pooled-OLS), random effects least squares (RE-OLS), fixed effects least squares (FE-OLS)) and dynamic panel (system

GMM model) estimation methods, to analyze the specific impact of inclusive finance on agricultural green development. The results show that there is a significant positive correlation between inclusive finance and agricultural green development, and inclusive finance plays an important role in promoting agricultural green development, expanding relevant research on the relationship between inclusive finance and agricultural green development. On this basis, some targeted countermeasures and suggestions are put forward, hoping to enrich the research perspectives of inclusive finance and agricultural green development and expand related research fields.

Some limitations of this paper are as follows: This paper studies the impact of inclusive finance on agricultural green development from a macro perspective, and the relationship between the two needs to be explored from a micro perspective in the future. The data in this paper belongs to the short panel, the analysis is short-term analysis, and further long-term analysis can be done in the future.

The chapters of this paper are arranged as follows. The first part is the introduction, which mainly introduces the research background and significance of this paper, the key issues, possible contributions and limitations of this paper. The second part is the literature review. The third part is the theoretical basis. The fourth part mainly includes development level of inclusive finance and agricultural green development level. The fifth part introduces the index system and model construction. The sixth part mainly includes conclusions and policy recommendations.

## Literature review

### Literature review on inclusive finance

The research on the connotation, characteristics, and measurement of inclusive finance has achieved rich results. The concept of inclusive finance was put forward by The United Nations in 2005, and it is believed that inclusive finance refers to the provision of appropriate and effective financial services for all groups of society at affordable costs. Some scholars select indicators from multiple dimensions to construct an indicator system for measuring the development level of inclusive finance. For example, Beck T, et al. (2007) propose indicators to measure financial inclusion [1]; Mandira Sarma (2008) adopts three dimensions of bank penetration, financial service accessibility, and financial service use utility [2]; Arora (2010) adopts three dimensions of service scope, convenience and use cost [3]; Gupte (2012) adopts four dimensions of financial penetration, financial usability, financial transaction convenience and financial transaction cost [4]; Yaoqun Zhang, et al. (2021) adopt three dimensions of financial services geographic penetration, availability, usage of utility [5]; Wenwen Wang (2021) uses three dimensions of geographic penetration, financial product accessibility, and financial service utility [6].

### Literature review on agricultural green development

The research on the connotation, characteristics, influencing factors, and measurement of agricultural green development have achieved rich results. Steven Haggblade, et al. (1989) believe that the difficulty and core of green agricultural development are the innovation of green agricultural development path and green technology innovation [7]. Unep (2011) believes that agricultural green development includes resource conservation, good ecological environment and long-term coordinated agricultural development, and has modern production methods [8]. Parviz Koohafkan, et al. (2012) believe that agricultural green development can help improve the safety and quality of agricultural products, optimize the ecological environment, and enhance the brand influence of agricultural products [9]. Adnan et al. (2019)

believes that agricultural green development is affected by the individual characteristics of farmers, family characteristics, farmers' cognition, and external environmental characteristics [10]. Qi Wei, et al. (2018) evaluate China's agricultural green development level from four dimensions: resource conservation, environmental friendliness, ecological conservation and quality and efficiency [11]. Hui Jie Zhao, et al. (2019) use the entropy weight method to evaluate the agricultural green development level of 13 provinces in China's main grain producing areas from four dimensions: resource conservation, environmental friendliness, high output efficiency, and life security [12].

## Literature review on the impact of inclusive finance on the agricultural green development

At present, scholars have less research on the impact of inclusive finance on agricultural green development, and more research is concentrated in the aspects of inclusive finance on rural residents' income, consumption, reduced urban and rural gaps, and promoting industrial structure upgrades (Bruhn, 2014) [13]. Kinnon (1973) proposes financial suppression in developing countries, and the generation of inclusive finance can effectively alleviate this phenomenon [14]. Dollar (2002) finds that rural financial development is one of the important factors to promote poverty reduction in rural areas [15]. Geoda et al. (2006) find that inclusive finance can significantly improve the income level of rural areas and lower than the median groups of income through credit, savings and other expanded financial services [16]. Lili Jiang et al. (2019) analyze the specific impact of inclusive finance on farmers' entrepreneurship [17]. Wei Zhang, et al. (2019) analyze the financial inducement mechanism to promote agricultural green development from three aspects: loan support, insurance protection and price protection [18]. Changying Hua (2022) analyzes the impact of digital inclusive finance on agricultural green development [19].

To sum up, some scholars have carried out relevant researches on inclusive finance and agricultural green development, but few literatures have studied the relationship between inclusive finance and agricultural green development. There is a lack of empirical research on the impact of inclusive finance on agricultural green development. Therefore, the main purpose of this study is to deeply analyze the internal relationship between inclusive finance and agricultural green development, to analyze the influence mechanism of inclusive finance on agricultural green development, and to conduct empirical research on the specific impact of inclusive finance on agricultural green development.

## Theoretical analysis of the impact of inclusive finance on agricultural green development

### Financial exclusion theory and inclusive finance theory

The concept of financial exclusion was proposed in the late 20th century. In order to chase interests to maximize interests, financial institutions have reduced or even cancel financial institutions in underdeveloped areas, making it difficult for group groups to obtain financial services. In the financial market, financial service structure is unbalanced, that is, the contradiction between the demand for financial funds and the supply of financial institutions with the supply of financial funds, which is one of the main manifestations of market failure (Kempson, Whyley, 1999) [20]. Guojun Zhang et al. (2014) believe that financial exclusion provides differentiated financial services to different social populations, which will exacerbate social unfairness and make the social functions of finance cannot be realized [21].

Helms (2006) proposes to change the problem of financial exclusion in the traditional financial system, especially for customers who have been excluded by the formal financial

system to obtain financial services, which can be said to be the direct purpose of developing inclusive finance [22]. Sarma & Pais (2011) believes that inclusive finance is opposite to finance, and inclusive finance means that all economic entities can enjoy financial products and services at reasonable prices fairly [23]. Jinpu Jiao (2006) believes that the inclusive financial system is based on the pursuit of sustainable and provides financial services including savings, credit, insurance and other financial services to all strata, including disadvantaged groups that are excluded from traditional financial systems outside the traditional financial system [24].

## Agricultural green development theory

Agricultural green development is a large agricultural in the broad sense, which is the integration and supplement of ecological agriculture, organic agriculture and natural agriculture. The relevant theories of agricultural green development can be traced back to the sustainable development theory. Both the sustainable development theory and the green growth theory originated from the Western academic circles. Gro, et al. (1987) propose *sustainable development* in *Our Common Future* [25]. FB Gomes, et al. (2001) propose *Sustainable Europe makes the world a better place*: *EU strategy for sustainable development* [26].

## Analysis of the mechanism of inclusive finance affecting agricultural green development

Inclusive finance can promote agricultural green development by improving the coverage and availability of financial services in rural areas, providing credit services, and providing insurance services.

**Inclusive finance can promote green agricultural development by increasing the coverage and availability of financial services in rural areas.** Inclusive finance can improve the coverage and availability of financial services in rural areas and reduce the cost of financial services, so it can better meet the financial needs of green agricultural business entities. Therefore, inclusive finance can promote agricultural green development. In the process of agricultural green development, there are prominent financial needs such as capital needs and insurance needs. The development of inclusive finance can effectively improve the coverage and availability of financial services by increasing the financial infrastructure such as ATM machines, bank card withdrawal points, and financial service points in rural areas, and by increasing the number of financial service personnel in rural areas. In addition, with the promotion and application of new technologies such as communications and networks, telephone banking, online banking, and e-banking have further popularized and developed in rural areas, which can further improve the coverage and availability of financial services in rural areas; and reduce the cost of providing financial services and improve the convenience and safety of providing financial services. Therefore, the development of inclusive finance can better meet the financial needs of agricultural green development.

**Inclusive finance can promote agricultural green development by providing credit services.** By providing moderate-cost credit services, inclusive finance can meet the capital needs generated in the process of agricultural green development, thereby promoting green agricultural development. On the one hand, in the process of promoting agricultural green development, it is necessary to take measures such as improving the agricultural production environment and promoting agricultural green production technology, which will generate a large demand for funds. On the other hand, green agricultural business entities have prominent capital needs in the process of promoting green agricultural development. The development of inclusive finance can better meet the capital needs generated in the process of agricultural green development. By increasing the branches or business outlets of financial

institutions operating in rural areas, and by increasing the number of financial service personnel in rural areas, inclusive finance can timely understand the financial needs of different green agricultural business entities, so that financial institutions can design and develop some new, flexible credit products, which can promote agricultural green development.

**Inclusive finance can promote agricultural green development by providing insurance services.** Inclusive insurance has the functions of transferring risks and dispersing risks, which is conducive to effectively dealing with natural risks, market risks and other risks in the process of agricultural green development, thereby promoting green agricultural development. In particular, agricultural insurance is conducive to dispersing and transferring the various risks faced by green agricultural business entities in the process of promoting agricultural green development. Agricultural insurance provides a guarantee for green agricultural business entities by providing compensation for losses caused by risks timely, and helps green agricultural business entities to resume production as soon as possible. Therefore, the development of inclusive insurance and agricultural insurance plays a very important role in promoting agricultural green development.

## Measurement of the development level of inclusive finance and agricultural green development level of 13provinces in China's main grain producing areas

On the basis of analyzing the interaction between agricultural green development and inclusive finance, this paper constructs a scientific and reasonable evaluation index system for the development level of inclusive finance to evaluate the development level of inclusive finance in 13 provinces in the main grain producing areas from 2011 to 2019. This paper also constructs an evaluation index system for the level of agricultural green development to measure the agricultural green development level of 13 provinces in the main grain producing areas from 2011 to 2019.

### Measurement of the development level of inclusive finance in 13 provinces in China's main grain producing areas

**Constructing an evaluation index system for the development level of inclusive finance.** Based on the previous research results and combined with the actual situation of the 13 provinces in China's main grain producing areas, this paper selects 8 indicators from the three dimensions of the geographic penetration of financial services, the availability of financial services, and the utility of financial services to measure the development level of inclusive finance in 13 provinces. The specific indicators are shown in **Table 1**.

**Calculation method of Index of Financial Inclusion.** Determining the weight of each indicator.

When assigning values to each index in the evaluation index system of the development level of inclusive finance, this paper adopts the coefficient of variation method to determine the weight of each index in the evaluation index system. The details are as follows: first calculate the average $\bar{X}_i$ (i = 1,2,...8) and standard deviation of each index $s_i$, then calculate the coefficient of variation of the index according to the formula $V_i = \frac{s_i}{\bar{X}_i}$, and then calculate the weight of the indicator $w_i$ according to $w_i = V_i / \sum_{i=1}^{8} V_i$

*Constructing an Index of Financial Inclusion.* Based on the established evaluation index system for the development level of inclusive finance, the Index of Financial Inclusion (IFI) was calculated to evaluate the development level of inclusive finance in 13 provinces.

Since the measurement units and economic meanings of the above 8 specific indicators are different, they cannot be directly compared with each other, so they need to be normalized.

**Table 1. Evaluation index system for the development level of inclusive finance.**

| Dimension Index | Specific indicators | Unit of measurem-ent | Index Measurement | Attributes |
|---|---|---|---|---|
| **Geographic penetration of financial services** | Number of banking financial institutions per 10,000 square kilometers | piece | Number of banking financial institutions/total land area | Positive indicator |
| | Number of employees in banking financial institutions per 10,000 square kilometers | piece | Number of employees in banking financial institutions/total land area | Positive indicator |
| **Availability of financial services** | Number of banking financial institutions per 10,000 people | piece | Number of banking financial institutions /number of permanent residents | Positive indicator |
| | Number of employees in banking financial institutions per 10,000 people | piece | Number of employees in banking financial institutions/number of resident population | Positive indicator |
| **Utility of financial services** | Deposit balance as a percentage of GDP | % | Deposit balance of financial institutions/GDP | Positive indicator |
| | Loan balance as a percentage of GDP | % | Loan balance of financial institutions/GDP | Positive indicator |
| | Insurance density | RMB10,000/ person | Original premium income /population | Positive indicator |
| | Insurance depth | % | Original premium income/GDP | Positive indicator |

The specific formula is: $d_i = w_i * \frac{(A_i - \min A_i)}{(\max A_i - \min A_i)}$, where $d_i$ represents the dimensionless measurement value of the i-th indicator, $w_i$ is the weight of the i-th indicator, $A_i$ is the original value of the i-th indicator, $\min A_i$ is the minimum values of the i-th indicator, $\max A_i$ is the maximum values of the i-th indicator. Since $0 \leqq w_i \leqq 1$, $0 \leqq d_i \leqq w_i$, that is, the larger the $d_i$, the greater the impact of this indicator on inclusive finance.

Referring to the research of Sarma (2008), the formula for calculating the Index of Financial Inclusion (IFI):

$$IFI = 1 - \frac{\sqrt{\sum (w_i - d_i)^2}}{\sqrt{\sum w_i^2}} \tag{1}$$

According to Formula (1), it can be seen that the value range of the Index of Financial Inclusion (IFI) is in [0,1]. When IFI = 0, the development of inclusive finance is the lowest; the closer IFI is to 1, the higher the development of inclusive finance; when IFI = 1, the development of inclusive finance is the highest.

*Data sources*. The data related to the development of inclusive finance mainly come from the Statistical Yearbook, the Statistical Bulletin of National Economic and Social Development, and the China Financial Yearbook of the 13 provinces in the main grain producing areas from 2012 to 2020. *China Regional Financial Operation Report* issued by the People's Bank of China, China Rural Financial Statistical Yearbook, and the website of the National Bureau of Statistics. The indicators in **Table 1** are used to calculate the Index of Financial Inclusion of 13 provinces.

*Calculation results of Index of Financial Inclusion in 13 provinces in China's main grain producing areas*. According to the above Formula (1), Index of Financial Inclusion of 13 provinces in China's main grain producing areas from 2011 to 2019 is calculated (**Table 2**). The higher Index of Financial Inclusion, the higher the development of inclusive finance. From 2011 to 2019, the overall development level of inclusive finance in 13 provinces has increased significantly, which is related to the emphasis on rural areas by provincial governments and financial departments in major grain-producing areas and increased financial investment. Among them, the provinces with high Index of Financial Inclusion scores are Jiangsu, Shandong and

**Table 2. Index of Financial Inclusion of 13 provinces in China's major grain producing areas from 2011 to 2019.**

| year<br>province | 2011 | 2012 | 2013 | 2014 | 2015 | 2016 | 2017 | 2018 | 2019 |
|---|---|---|---|---|---|---|---|---|---|
| Hebei | 0.2828 | 0.2938 | 0.3159 | 0.3373 | 0.3788 | 0.4142 | 0.4414 | 0.4526 | 0.4651 |
| Inner Mongolia | 0.0869 | 0.0957 | 0.0993 | 0.1106 | 0.1330 | 0.1596 | 0.1727 | 0.1804 | 0.1874 |
| Liaoning | 0.3454 | 0.3358 | 0.4175 | 0.4154 | 0.4602 | 0.4851 | 0.5004 | 0.4817 | 0.4969 |
| Jilin | 0.1871 | 0.1899 | 0.2036 | 0.2326 | 0.2747 | 0.3167 | 0.3405 | 0.3242 | 0.3371 |
| Heilongjiang | 0.1304 | 0.1407 | 0.1537 | 0.1856 | 0.2146 | 0.2322 | 0.2605 | 0.2575 | 0.2624 |
| Jiangsu | 0.4955 | 0.5184 | 0.5407 | 0.5713 | 0.6129 | 0.6734 | 0.7261 | 0.7013 | 0.7275 |
| Anhui | 0.2400 | 0.2276 | 0.2574 | 0.2769 | 0.3213 | 0.3367 | 0.3617 | 0.3637 | 0.3766 |
| Jiangxi | 0.1689 | 0.1783 | 0.1940 | 0.2193 | 0.2520 | 0.2762 | 0.2925 | 0.2906 | 0.2996 |
| Shandong | 0.3719 | 0.3878 | 0.4320 | 0.4379 | 0.4499 | 0.5153 | 0.5412 | 0.5497 | 0.5605 |
| Henan | 0.2999 | 0.3132 | 0.3297 | 0.3349 | 0.3744 | 0.4080 | 0.4828 | 0.4499 | 0.4553 |
| Hubei | 0.1972 | 0.2035 | 0.2174 | 0.2332 | 0.2562 | 0.1512 | 0.3080 | 0.3103 | 0.3330 |
| Hunan | 0.1765 | 0.1814 | 0.1892 | 0.2032 | 0.2271 | 0.2540 | 0.2776 | 0.2912 | 0.3008 |
| Sichuan | 0.1987 | 0.2052 | 0.2179 | 0.2353 | 0.2558 | 0.2876 | 0.2958 | 0.3016 | 0.2981 |

Liaoning; the provinces with low Index of Financial Inclusion scores are Inner Mongolia, Heilongjiang, Sichuan, Jiangxi, and Hunan; the provinces with middle Index of Financial Inclusion scores are Hebei, Henan, Anhui, Jilin, and Hubei.

**Fig 1** shows the specific situation of the Index of Financial Inclusion of 13 provinces in China's main grain producing areas in 2019. It can be seen that there are large differences in the inclusive finance development indices of the 13 provinces in China's main grain producing areas. Among them, Jiangsu Province has the highest score of Index of Financial Inclusion, which is 0.7275; Inner Mongolia has the lowest score of Index of Financial Inclusion, which is 0.1874.

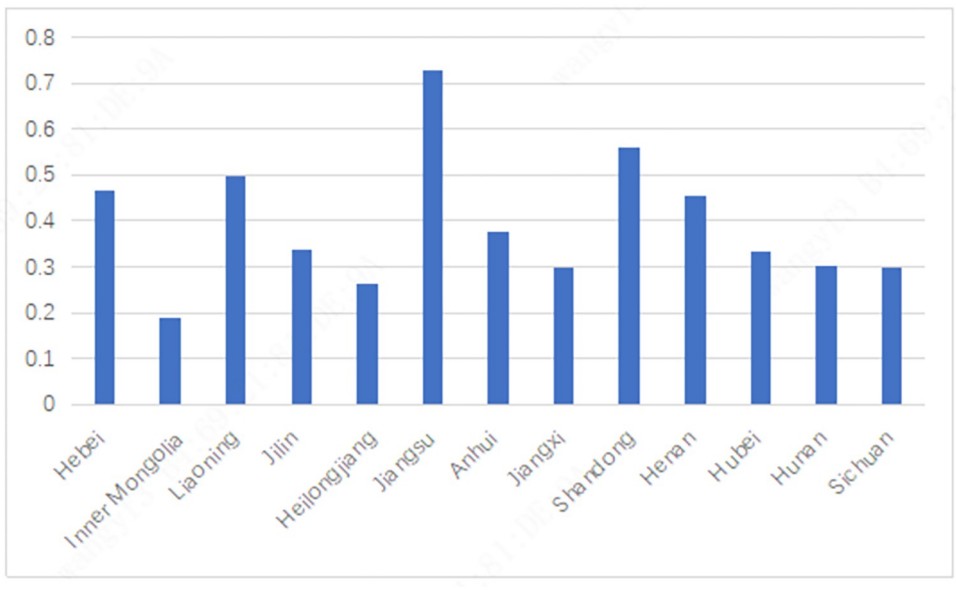

**Fig 1. Index of Financial Inclusion of 13 provinces in China' s major grain producing areas in 2019.**

## Measurement of agricultural green development level in 13 provinces in China's main grain producing areas

**Evaluation index system for agricultural green development level.** The elements of agricultural green development are mainly reflected in the aspects of resources, environment and quality efficiency. Based on the previous research results of relevant scholars, combined with the situation of agricultural green development in 13 provinces in China's main grain producing areas and the availability of data, this paper focuses on four dimensions: resource conservation, environmental friendliness, quality and efficiency improvement, and life security. Twelve specific indicators are selected as the evaluation index system of agricultural green development level in 13 provinces in China's main grain producing areas. The specific indicators are shown in **Table 3**. Resource conservation is the basic feature of agricultural green development, emphasizing the improvement of the utilization rate of cultivated land and water resources. In this paper, two specific indicators, namely the multi-cropping index of arable land and agricultural water consumption per gross agricultural output were selected to evaluate the resource conservation of agricultural green development in 13 provinces. The essence of environmental friendliness is the harmonious unity of agricultural development and resource environmental protection, emphasizing the protection of resource bases for agricultural production. In this paper, four indicators of pesticide application intensity, chemical fertilizer application intensity, agricultural film use intensity, and forest coverage rate were selected to evaluate the environmental friendliness of agricultural green development in 13 provinces. The improvement of quality and efficiency is the direct goal of green agricultural development, emphasizing both the quantity and quality of output. In this paper, five

**Table 3. Evaluation index system and weight of agricultural green development level.**

| Dimension Index | Specific indicators | Unit of measurement | Index Measurement | Attributes | Weight |
|---|---|---|---|---|---|
| **Resource conservation** | Multi-cropping index of arable land | - | Crop sown area/arable land area | Negative indicator | 0.0986 |
| | Agricultural water consumption per gross agricultural output | m$^3$ | Agricultural water consumption/gross agricultural output | Negative indicator | 0.0901 |
| **Environmental friendliness** | Pesticide application intensity | kg/ha | Pesticide application rate/sown area | Negative indicator | 0.0986 |
| | Chemical fertilizer application intensity | kg/ha | Fertilizer application rate/sown area | Negative indicator | 0.0986 |
| | Agricultural film use intensity | kg/ha | Agricultural film usage/sown area | Negative indicator | 0.0986 |
| | Forest coverage rate | % | China Environmental Statistical Yearbook | Positive indicator | 0.0986 |
| **Quality and efficiency improvement** | Number of green food label products per unit area | Piece/10,000 hectares | Green food label product quantity/arable land area | Positive indicator | 0.0986 |
| | Number of green enterprises per unit area | Piece/10,000 hectares | Number of green enterprises/arable land area | Positive indicator | 0.0986 |
| | Grain yield per unit area | kg/ha | Total grain production/grain sown area | Positive indicator | 0.0986 |
| | Total agricultural output value per sown area | RMB yuan 10,000/ha | Gross agricultural output/sown area | Positive indicator | 0.0986 |
| | Per capita agricultural, forestry, animal husbandry and fishery output value | RMB yuan 10,000/person | Gross output value of agriculture, forestry, animal husbandry and fishery/employees of agriculture, forestry, animal husbandry and fishery | Positive indicator | 0.0986 |
| Life security | Per capita disposable income of rural residents | RMB yuan | Year Statistical Yearbook | Positive indicator | 0.0986 |

indicators are selected to measure the quantity, quality and efficiency of agricultural green output in 13 provinces, including the number of green food label products per unit area, the number of green enterprises per unit area, the grain yield per unit area, the total agricultural output value per sown area, and the per capita agricultural, forestry, animal husbandry and fishery output value. The goal of green agricultural development is to meet people's demand for high-quality and safe agricultural products, while also focusing on increasing farmers' income and ensuring farmers' lives. This paper chooses the per capita disposable income of rural residents as the main indicator to measure the living security.

**Data sources.** The relevant data on agricultural green development comes from the Statistical Yearbook, China Rural Statistical Yearbook, China Water Conservancy Yearbook, the National Bureau of Statistics, and the websites of China Green Food Development Center of 13 provinces in the main grain producing areas from 2012 to 2020.

**Determining indicator weights.** In this paper, the entropy weight method is used as the method to determine the index weight, and the entropy weight method can help obtain a more objective weight.

The first step is to standardize the data. The original data of all indicators are standardized by the dispersion standardization method, and 0.0001 was added to the standardized formula to avoid the meaningless assignment of zero values. There are k provinces, n years, and j indicators, then $x_{ikj}$ is the j-th indicator value of province k in the i-th year. The positive index data is standardized by Formula (2), and the negative index data is standardized by Formula (3).

Positive indicator normalization formula:

$$x'_{ikj} = \frac{x_{ikj} - x_{min}}{x_{max} - x_{min}} + 0.0001 \tag{2}$$

Negative indicator normalization formula:

$$x'_{ikj} = \frac{x_{max} - x_{ikj}}{x_{max} - x_{min}} + 0.0001 \tag{3}$$

Among them, $x'_{ikj}$ is the indicator data after standardization, $x_{ikj}$ is the original data, $x_{min}$ represents the minimum value of the indicator, and $x_{max}$ represents the maximum value of the indicator.

The second step is to determine the indicator weights:

$$Y_{ikj} = \frac{x'_{ikj}}{\sum_i \sum'_{ikj}} \tag{4}$$

The third step is to calculate the entropy value of the j-th indicator:

$$E_j = -r\sum_i \sum_k Y_{ikj} ln(Y_{ikj}), \quad r = ln(in) \tag{5}$$

The fourth step is to calculate the difference coefficient of the j-th indicator:

$$G_j = 1 - E_j \tag{6}$$

The fifth step is to calculate the weight of each indicator:

$$W_j = \frac{G_j}{\sum_j G_j} \tag{7}$$

The sixth step is to calculate the comprehensive score of the agricultural green development level of each province:

$$H_{ik} = \sum_j W_j x'_{ikj} \tag{8}$$

**Analysis of the calculation results of the Agricultural Green Development Index of 13 provinces in China's main grain producing areas.** The calculation results of the Agricultural Green Development Index of the 13 provinces in China's main grain producing areas from 2011 to 2019 are shown in **Table 4**. The higher the Agricultural Green Development Index, the higher the degree of agricultural green development. From 2011 to 2019, the level of agricultural green development in 13 provinces has been greatly improved. Among them, the provinces with high Agricultural Green Development Index scores are Jiangsu, Heilongjiang, Shandong, Liaoning and Hubei; the provinces with low Agricultural Green Development Index scores are Inner Mongolia, Hebei, Anhui, and Henan; the provinces with middle Agricultural Green Development Index scores are Sichuan, Jilin, Hunan, and Jiangxi.

## Index system and model construction

### Variable

**The explained variable.** This paper uses Agricultural Green Development Index (agi) from 2011 to 2019 to measure the agricultural green development level of the 13 provinces in China's main grain-producing areas. The relevant data are derived from the above calculation results.

**The core explanatory variables.** This paper uses the 2011–2019 Index of Financial Inclusion (IFI) to measure the development level of inclusive finance in 13 provinces in China's main grain producing areas. The relevant data are derived from the above calculation results.

**The control variables.** In order to more accurately measure the promotion effect of inclusive finance on agricultural green development and minimize errors caused by other factors, this paper selects three representative indicators as control variables. The original data of all control variables were obtained from the website of the National Bureau of Statistics. The specific control variables are as follows: regional economic growth level (ln *gdp*), measured by the logarithm of real GDP per capita; industrial structure (*is*), measured by the proportion of the

**Table 4. Agricultural Green Development Index of 13 provinces in China's main grain producing areas from 2011 to 2019.**

| year / province | 2011 | 2012 | 2013 | 2014 | 2015 | 2016 | 2017 | 2018 | 2019 | mean | rank |
|---|---|---|---|---|---|---|---|---|---|---|---|
| Hebei | 0.3903 | 0.4166 | 0.4342 | 0.4279 | 0.4257 | 0.4079 | 0.4239 | 0.4542 | 0.4741 | 0.4283 | 11 |
| Inner Mongolia | 0.4167 | 0.4059 | 0.4159 | 0.4193 | 0.4124 | 0.4238 | 0.4235 | 0.4581 | 0.4838 | 0.4288 | 10 |
| Liaoning | 0.4429 | 0.4487 | 0.4736 | 0.4730 | 0.4896 | 0.4813 | 0.4941 | 0.5053 | 0.5318 | 0.4822 | 4 |
| Jilin | 0.4385 | 0.4469 | 0.4562 | 0.4586 | 0.4588 | 0.4417 | 0.4476 | 0.4590 | 0.4815 | 0.4543 | 7 |
| Heilongjiang | 0.4564 | 0.4652 | 0.4877 | 0.5010 | 0.5039 | 0.5229 | 0.4972 | 0.5628 | 0.6132 | 0.5123 | 2 |
| Jiangsu | 0.4718 | 0.4632 | 0.4812 | 0.5156 | 0.5012 | 0.5232 | 0.5397 | 0.5774 | 0.6270 | 0.5223 | 1 |
| Anhui | 0.3508 | 0.3582 | 0.3710 | 0.3965 | 0.3866 | 0.4053 | 0.4934 | 0.4682 | 0.5016 | 0.4146 | 12 |
| Jiangxi | 0.4064 | 0.3962 | 0.4063 | 0.4195 | 0.4158 | 0.4433 | 0.4519 | 0.4721 | 0.4930 | 0.4338 | 9 |
| Shandong | 0.4304 | 0.4350 | 0.4704 | 0.5001 | 0.4883 | 0.4832 | 0.4981 | 0.5259 | 0.5505 | 0.4869 | 3 |
| Henan | 0.3585 | 0.3652 | 0.3751 | 0.4023 | 0.4055 | 0.4071 | 0.4239 | 0.4496 | 0.4757 | 0.4070 | 13 |
| Hubei | 0.4264 | 0.4236 | 0.4438 | 0.4594 | 0.4536 | 0.4730 | 0.4878 | 0.5039 | 0.5325 | 0.4671 | 5 |
| Hunan | 0.4265 | 0.4207 | 0.4434 | 0.4366 | 0.4380 | 0.4370 | 0.4530 | 0.4756 | 0.5380 | 0.4521 | 8 |
| Sichuan | 0.3999 | 0.4038 | 0.4214 | 0.4385 | 0.4355 | 0.4684 | 0.4830 | 0.5048 | 0.5363 | 0.4546 | 6 |

added value of agriculture, forestry, animal husbandry and fishery in GDP; the proportion of environmental protection expenditure (*ef*) is measured by the proportion of environmental protection expenditure in fiscal expenditure.

## Variable descriptive statistics

As can be seen from Table 5, the maximum and minimum values of each variable that there is a certain gap between each variable in different years. Among them, the average level of agricultural green development (agi) of the 13 provinces in China's main grain producing areas is 0.4573, the minimum value is 0.3508, and the maximum value is 0.6270, indicating that there is a certain gap in the green agricultural development level of the 13 provinces, but the gap is not particularly prominent. The average level of the Index of Financial Inclusion (*ifi*) of the 13 provinces in the main grain producing areas is 0.3213, the minimum value is 0.0869, and the maximum value is 0.7275, indicating that the level of financial inclusion in the 13 provinces is quite different. At the same time, the logarithm of real GDP per capita (ln *gdp*), the industrial structure (*is*), and the proportion of environmental protection expenditures (*ef*) in the 13 provinces in the main grain producing areas also have certain differences.

## Model setting

In order to better compare the rationality of the empirical analysis results, this paper simultaneously uses static panels (mixed least squares estimation (Pooled-OLS), random-effects least-squares estimation (RE-OLS), fixed-effects least squares estimation (FE- OLS)) estimation method for analysis. Considering that agricultural green development is a dynamic process with endogenous problems, this paper further establishes a systematic GMM model for research. The specific model is as follows.

Static panel regression model:

$$agi_{it} = \alpha + \beta_1 ifi_{it} + \beta_2 \ln gdp_{it} + \beta_3 is_{it} + \beta_4 ef_{it} + \mu_t + \lambda_{it} \tag{9}$$

Dynamic panel regression model:

$$agi_{it} = \alpha + \alpha_1 agi_{it-1} + \beta_1 ifi_{it} + \beta_2 \ln gdp_{it} + \beta_3 is_{it} + \beta_4 ef_{it} + u_t + \lambda_{it} \tag{10}$$

Among them, $agi_{it}$ is the Agricultural Green Development Index, which is the explained variable, $\alpha$ and $\beta$ represents the regression coefficient, $ifi_{it}$ is the Index of Financial Inclusion, ln $gdp_{it}$ is the logarithm of real GDP per capita, $is_{it}$ is the industrial structure, and $ef_{it}$ is the proportion of environmental protection expenditure.

## Regression results of the impact of inclusive finance on agricultural green development in 13 provinces in China's main grain producing areas

Table 6 shows the static panel regression analysis results (mixed least squares, fixed effects and random effects) and dynamic panel (system GMM) model regression analysis results of the

**Table 5. Descriptive statistics of variables.**

| Variable | Mean | Standard Deviation | Minimum | Maximum | Number of Samples |
|---|---|---|---|---|---|
| *agi* | 0.4573 | 0.0505 | 0.3508 | 0.6270 | 117 |
| *ifi* | 0.3213 | 0.1395 | 0.0869 | 0.7275 | 117 |
| **ln** *gdp* | 1.6026 | 0.3757 | 0.8863 | 2.4499 | 117 |
| *is* | 0.1006 | 0.0304 | 0.0431 | 0.2338 | 117 |
| *ef* | 0.0307 | 0.0109 | 0.0087 | 0.0613 | 117 |

**Table 6. Regression results of the impact of inclusive finance on agricultural green development in 13 provinces in China's main grain producing areas.**

| variable | Mixed Least Squares Estimation | fixed effects | (random effects) | system GMM |
|---|---|---|---|---|
| L.agi | | | | -0.309 (-0.85) |
| ifi | 0.200*** (6.72) | 0.146*** (2.69) | 0.154*** (3.66) | 0.275* (1.74) |
| lnrgdp | 0.136*** (16.26) | 0.153*** (7.63) | 0.143*** (9.17) | 0.262*** (2.80) |
| is | 1.585*** (20.31) | 1.010*** (4.83) | 1.031*** (5.92) | 4.148** (2.49) |
| ef | 0.452 (1.51) | 0.765** (2.01) | 0.691** (2.01) | 3.208 (1.38) |
| Constant term | | 0.040 (1.02) | 0.054 (1.57) | -0.435 (-1.40) |
| N | 117 | 117 | 117 | 104 |
| $R^2$ | 0.532 | 0.872 | 0.732 | |
| Arellano-Bond | | | | 0.344 |
| Hansen test | | | | 0.159 |

$p < 0.1$

**$p < 0.05$

***$p < 0.01$

impact of inclusive finance on agricultural green development in 13 provinces in China's main grain producing areas.

From the static panel regression analysis results, it can be seen that there is a significant positive correlation between the development level of inclusive finance and agricultural green development, indicating that the development of inclusive finance has a significant role in promoting green agricultural development. The control variables such as per capita real GDP, the proportion of the added value of agriculture, forestry, animal husbandry and fishery to GDP, and the proportion of environmental protection expenditure also have a positive impact on the green development of agriculture. This shows that the increase in the actual GDP per capita, the increase in the proportion of the added value of agriculture, forestry, animal husbandry and fishery to GDP, the increase in the proportion of environmental protection expenditure, which is conducive to promoting the agricultural green development. It can be seen from the results of dynamic panel regression analysis that after the residuals are tested, the p-values of Hansen test and Arellano-Bond in the system GMM estimation are both greater than 0.1, indicating that the model passes the correlation test, and the estimated empirical analysis result is effective. In the system GMM model, it can be seen that the development of inclusive finance has a positive impact on agricultural green development, and the control variables such as per capita real GDP and the added value of agriculture, forestry, animal husbandry and fishery to GDP also have a positive impact on agricultural green development.

Existing studies mainly focus on the impact of inclusive finance on urban and rural income, consumption, poverty reduction, employment, entrepreneurship and innovation. However, there is a lack of empirical research on the relationship between inclusive finance and agricultural green development. The results show that there is a significant positive correlation between the level of inclusive financial development, and inclusive finance plays an important role in promoting the green development of agriculture. The results of the study extends the existing studies in the field of finance and agricultural green development.

## Stability test

In order to test whether the above regression analysis is stable, this paper uses the transformation replacement method for stable testing. With the development of mobile information technology and digital technology, inclusive finance gradually combines with digital technology,

producing digital inclusive finance. In 2016, the G20 meeting formulated a high-level principle of digital inclusive finance, and digital inclusive finance gradually became an important part of inclusive finance. Therefore, this paper uses the Digital Inclusive Financial Index calculated by Peking University instead of the previously calculated Inclusive Financial Index for a stable test. It can be seen from the regression results (**Table 7**) that the impact of digital inclusive finance on agricultural green development is significantly positive in the mixed minimum daily estimation model, in the fixed effect model, in the random effect model, and in the system GMM model. The regression results are consistent with the regression results in **Table 6** above, indicating that the study of this paper has good stability, and inclusive finance has a significant role in promoting agricultural green development.

## Conclusions and policy recommendations

### Conclusions

This paper uses both static panel and dynamic panel (system GMM model) estimation methods to empirically analyze the specific impact of inclusive financial development on agricultural green development. The results show that there is a significant positive correlation between the level of inclusive financial development, real GDP per capita, the proportion of the added value of agriculture, forestry, animal husbandry and fishery in GDP, the proportion of environmental protection expenditure and agricultural green development. Therefore, this paper draws the following conclusions: improving the development level of inclusive finance is conducive to promoting agricultural green development; the increase in per capita real GDP is conducive to promoting agricultural green development; the greater the proportion of the added value of agriculture, forestry, animal husbandry and fishery in GDP, the more conducive to promoting agricultural green development; the increase in the proportion of environmental protection expenditure is conducive to promoting agricultural green development. Based on its findings, this paper proposes the following policy recommendations.

### Policy recommendations

**Developing inclusive finance to better promote agricultural green development.** Inclusive finance has a significant role in promoting agricultural green development. Therefore, it is

**Table 7. Results of stability regression analysis.**

| variable | Mixed Least Squares Estimation | fixed effects | (random effects) | system GMM |
|---|---|---|---|---|
| L.agi | | | | 0.619*** (2.630) |
| ifi | 0.021*** (4.548) | 0.021*** (3.676) | 0.022*** (5.365) | 0.017** (2.520) |
| lnrgdp | 0.149*** (16.932) | 0.088*** (2.853) | 0.081*** (3.838) | 0.032 (1.560) |
| is | 1.392*** (17.101) | 0.666*** (2.895) | 0.616*** (3.413) | 0.325 (1.651) |
| ef | 1.114 (3.054) | 1.502** (3.546) | 1.323*** (3.554) | 0.102 (0.232) |
| Constant term | | 0.166** (2.900) | 0.184*** (4.400) | 0.058 (1.151) |
| N | 117 | 117 | 117 | 104 |
| $R^2$ | 0.444 | 0.883 | 0.771 | |
| Arellano-Bond | | | | 0.347 |
| Hansen test | | | | 0.179 |

$p < 0.1$

**$p < 0.05$

***$p < 0.01$

necessary to vigorously develop inclusive finance to better promote agricultural green development. On the one hand, it is suggested to further improving the broad coverage of inclusive finance, further strengthen the construction of financial institutions' outlets, ATM machines, and cash-out points for farmers, and the construction of digital networks and other inclusive financial infrastructure in China's main grain producing areas. At the same time, increasing the number of financial practitioners in the rural areas of China's main grain producing areas can provide better financial services for promoting green agricultural development. On the other hand, it is better to further optimize inclusive financial products in China's major grain producing areas. On the basis of further understanding the types of financial needs of green agricultural operators and their characteristics, rural financial institutions can optimize the design of inclusive green credit products and inclusive green insurance products with flexible terms, flexible quotas, diverse borrowing methods, and moderate costs. Inclusive green financial products can help to better meet the capital needs and risk-transferring needs of green agricultural business entities. Therefore, agricultural green development can be promoted by providing better financial support and services.

## Promoting economic growth to better promote agricultural green development

Economic growth plays a significant role in promoting agricultural green development. Therefore, it is necessary to vigorously promote the economic growth of China's major grain producing areas to better promote agricultural green development. It is necessary to vigorously develop local leading industries and advantageous industries to promote economic growth according to the resource endowments and advantageous resources of each province. Economic growth can improve people's income level, increase people's demand for high-quality agricultural products, and enhance people's purchasing power for high-quality agricultural products, which is conducive to promoting agricultural green development.

**Promoting the development of agriculture, forestry, animal husbandry and fishery to better promote agricultural green development.** Agriculture, forestry, animal husbandry and fishery have a significant role in promoting agricultural green development. Therefore, it is necessary to further promote the development of agriculture, forestry, animal husbandry and fishery in China's main grain producing areas to increase the proportion of agriculture, forestry, animal husbandry and fishery in GDP to better promote agricultural green development. In China's main grain producing areas, the development of agriculture, forestry, animal husbandry and fishery is more important, which is not only related to food security, but also to the long-term stability of the country. Therefore, it is necessary to increase the investment in agriculture, forestry, animal husbandry and fishery in China's main grain producing areas. More attention should be paid to the green development of agriculture, forestry, animal husbandry and fishery.

**Increasing environmental protection expenditures to better promote agricultural green development.** Environmental protection expenditure has a significant role in promoting agricultural green development. Therefore, it is necessary to further increase the proportion of environmental protection expenditure in total fiscal expenditure to better promote agricultural green development. The environmental protection expenditure in China's main grain producing areas should be increased, and the increase in the proportion of environmental protection expenditure in various regions should be higher than the increase in the proportion of local financial expenditure, which is conducive to further improving the ecological environment of the main grain producing areas, optimizing the ecological environment for agricultural green development, promoting agricultural green development.

## Supporting information

**S1 Data.**
(ZIP)

## Author Contributions

**Data curation:** Lili Jiang.

**Methodology:** Yufan Ru.

**Software:** Zhifei Hu.

**Writing – original draft:** Aihua Tong.

**Writing – review & editing:** Zhongrong Xu, Yifeng Wang.

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
