## [Decision Letter · Decision Letter 0]

5 Aug 2022

PONE-D-22-20221Research on the Impact of Inclusive Finance on Agricultural Green Development——Empirical Analysis of China’s Main Grain Producing AreasPLOS ONE

Dear Dr. Yifeng Wang,

Thank you for submitting your manuscript to PLOS ONE. After careful consideration, we feel that it has merit but does not fully meet PLOS ONE’s publication criteria as it currently stands. Therefore, we invite you to submit a revised version of the manuscript that addresses the points raised during the review process.

We look forward to receiving your revised manuscript.

Kind regards,

María del Carmen Valls Martínez, Ph.D.

Academic Editor

PLOS ONE

Journal Requirements:

The name of the colleague or the details of the professional service that edited your manuscript.

A copy of your manuscript showing your changes by either highlighting them or using track changes.(uploaded as a *supporting information* file)

A clean copy of the edited manuscript. (uploaded as the new *manuscript* file)

Reviewers' comments:

Reviewer's Responses to Questions

**Comments to the Author**

1. Is the manuscript technically sound, and do the data support the conclusions?

Reviewer #1: Yes

Reviewer #2: Yes

2. Has the statistical analysis been performed appropriately and rigorously? 

Reviewer #1: Yes

Reviewer #2: Yes

3. Have the authors made all data underlying the findings in their manuscript fully available?

Reviewer #1: No

Reviewer #2: Yes

4. Is the manuscript presented in an intelligible fashion and written in standard English?

Reviewer #1: No

Reviewer #2: Yes

5. Review Comments to the Author

Reviewer #1: The paper " Research on the impact of inclusive finance on agricultural green development—Empirical analysis of China’s main grain producing areas" is interesting for journal readers. Kindly take note of the following specific comments to make it better.

1. A good introduction consists of the importance of the topic, why this research is significant, what is known in the literature and what is unknown, what is the research problem, what theoretical framework has been used, what are the main findings, and how it contributes to the literature.

2. The purpose of the research, the research gap, and the research field significance should be clearly stated. The current state of the art should be reviewed, and critical publications must be cited. You did not explain what this gap is. The same problem is with the aim of the paper.

3. The novelty needs to be strengthened.

4. The literature review section is too long and less international paper citations. The literature review stands irrelevant the authors need to develop a conceptual and theoretical framework but a comprehensive one.

5. The conceptual framework is weak; the authors need to define a proper theoretical framework under which this study has been conducted. The theoretical background is not clear as this part's central theme of this part and why is this title not precisely stated. You need to be more clear about how theory arguments affect the constructs revealing the scientific relevance and contribution of your work.

6. Please complete the information on how the results were validated. The results section is worth expanding. How do they correspond with other research on the topic?

7. The authors can also show how this study differs from other studies published in the PLOS One journal.

8. By the way, I would recommend that English writing should be proofread since there are lots of grammatical and technical errors throughout this paper.

Reviewer #2: 1. English Proofreading should be done.

2. The gap in the study should be elaborated. Why does this study need to be conducted?

3. Compare the results of the study with the existing studies in the field.

4. Conclusion and policy recommendations should be separately written.

5. Limitations of the study should be given.

6. PLOS authors have the option to publish the peer review history of their article (what does this mean?). If published, this will include your full peer review and any attached files.

Reviewer #1: **Yes: **RONI BHOWMIK

Reviewer #2: No

---

## [Author Response · Author response to Decision Letter 0]

22 Aug 2022

We have modified the maunscript according to the reviewers' comment.

---

## [Decision Letter · Decision Letter 1]

30 Aug 2022

Research on the Impact of Inclusive Finance on Agricultural Green Development: Empirical Analysis of China’s Main Grain Producing Areas

PONE-D-22-20221R1

Dear Dr. Yifeng Wang,

We’re pleased to inform you that your manuscript has been judged scientifically suitable for publication and will be formally accepted for publication once it meets all outstanding technical requirements.

Kind regards,

María del Carmen Valls Martínez, Ph.D.

Academic Editor

PLOS ONE

Reviewers' comments:

Reviewer's Responses to Questions

**Comments to the Author**

1. If the authors have adequately addressed your comments raised in a previous round of review and you feel that this manuscript is now acceptable for publication, you may indicate that here to bypass the “Comments to the Author” section, enter your conflict of interest statement in the “Confidential to Editor” section, and submit your "Accept" recommendation.

Reviewer #1: All comments have been addressed

Reviewer #2: All comments have been addressed

2. Is the manuscript technically sound, and do the data support the conclusions?

Reviewer #1: Yes

Reviewer #2: Yes

3. Has the statistical analysis been performed appropriately and rigorously? 

Reviewer #1: Yes

Reviewer #2: Yes

4. Have the authors made all data underlying the findings in their manuscript fully available?

Reviewer #1: Yes

Reviewer #2: Yes

5. Is the manuscript presented in an intelligible fashion and written in standard English?

Reviewer #1: Yes

Reviewer #2: Yes

6. Review Comments to the Author

Reviewer #1: (No Response)

Reviewer #2: The current form of the manuscript meets the requirements of an academic paper. So it is accepted in the current form.

7. PLOS authors have the option to publish the peer review history of their article (what does this mean?). If published, this will include your full peer review and any attached files.

Reviewer #1: **Yes: **Dr. RONI BHOWMIK

Reviewer #2: No

---

## [Editor Report · Acceptance letter]

21 Sep 2022

PONE-D-22-20221R1 

Research on the Impact of Inclusive Finance on Agricultural Green Development：Empirical Analysis of China’s Main Grain Producing Areas 

Dear Dr. Wang:

I'm pleased to inform you that your manuscript has been deemed suitable for publication in PLOS ONE. Congratulations! Your manuscript is now with our production department. 

Kind regards, 

on behalf of

Dr. María del Carmen Valls Martínez 

Academic Editor

PLOS ONE